# An ECCD—Electronic Charge Compensation Device—As a Quantum Dissipative System

**Eusebio Bernabeu** [1,*], **Javier Maldonado** [2] **and María A. Sáenz-Nuño** [3]

[1] Instituto Universitario de Ciencias Ambientales, Universidad Complutense de Madrid Ciudad Universitaria, 28040 Madrid, Spain

[2] Dinnteco International, C/Pere d'Urgel, 10, AD500 Andorra la Vella, Principado de Andorra; DINNTECO Factory Gasteiz, SL. C/Aibarra, 30. pabellón, 1. 01010 Vitoria, Spain; jmaldonado@dinnteco.com

[3] ICAI, Escuela Técnica Superior de Ingeniería & IIT, Universidad Pontificia Comillas de Madrid, C/Alberto Aguilera, 25, 28015 Madrid, Spain; msaenz@comillas.edu

\* Correspondence: ebernabeu@fis.ucm.es; Tel.: +34-607-836-502



**Featured Application: An ECCD is a passive device of security that prevents impacts of lightning, protects infrastructure from disturbances of radiofrequency, and protects against derivative exogenous electric current pulses.**

**Abstract:** An electronic charge compensation device (ECCD) is a passive device that carries electrical currents away, on time, to the electrical Earth field. It prevents lightning's impacts, derivative electric current pulses, and reduces the radiofrequency disturbances in the protected area. The objective of this paper is to give a physical explanation of the operation of an ECCD's performance and advantages. The operation of an ECCD is the result of two actions: the static electric field and the evanescent and resonant electrical radiofrequency field in the nearby external adjoining to dielectric-metal zone of ECCD. The energy absorption only is logically justified considering a super-absorption process as an end of chain of resonant quantum event. In this study, a multi-resonant process was inferred from an exhaustive radiofrequency simulation analysis made on an ECCD. The primary experiment was a long-time-frame statistical analysis of seven different, real stations. Those empirical results were derived from real METEORAGE environmental services data. Finally, a prospective for new applications is given.

**Keywords:** security; radiofrequency electromagnetic protection; dissipative quantum system at RF domain

---

## 1. Introduction

An electronic charge compensation device ECCD is a passive device that carries electrical currents, in time, to the electrical Earth field. The device we examined is a patent of DINNTECO International (P1); it is a security protector —DDCE, dispositivo de compensación de cargas eléctricas— that prevents lightning's impacts to infrastructure (telecommunication towers, sportive buildings, civil and heritage constructions, factories, ports, and airports, among others). The protection also includes derivative exogenous electric current pulses—generated by distant impact lightning—and reduces the radiofrequency disturbances in the protected area. This radiofrequency perturbation can have its origins in cloud–cloud lightning, in the emission of near telecommunication towers, in own friction between clouds by the wind and other atmospheric electrical changes, or by occasional external incidents. Obviously, the protection covers people, animals, electronic instruments, and all belongings in a wide coverage area (radius of $\cong$ 200 m).

It is quite convenient to start with a presentation of the statistical results of lightning's impacts around an ECCD, from 16 years ago until now. That related to empirical analysis: Section 2. The ECCD quantum description came up as consequence of the numerical simulation analysis developed at the INTA (Instituto Nacional de Técnica Aeroespacial) of Spain to explore the radiofrequency interactions with an ECCD; in particular, with an exhaustive study of two models of ECCD produced by DINNTECO: DDCE 100 and DDCE 50 (SM 2). Section 3 shows the most significant results focused only on DDCE 100 model (see Figure 1), because that is the extensively used one, but the results are also applicable to similar systems. Finally, Section 4 show and justifies the different quantum implications that possibly involve the basic support of an ECCD.

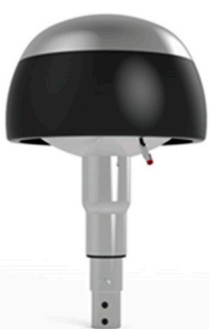

**Figure 1.** Photography of an electronic charge compensation device (ECCD)—DINNTECO 100 model.

## 2. Spatial Distribution of Lightning Impacts around the ECCD: A Statistical Analysis and an Empirical Rule on Protection Efficiency

The protection efficiency of the ECCD is by now supported by the test plan that DINNTECO International has been carrying out systematically and continuously for years in installations with a high risk of lightning impact, high lightning density, and high "ceraunic" activity. At all the facilities where these real tests are carried out, the condition that all their metallic parts should be at the same electrical potential as the lower hemisphere of the DDCE is fulfilled. These tests visualize the lightning activity detected within 2 km around the protected installation. The tests are based on data provided by an external company of recognized prestige: METEORAGE [1] (SM 2), whose lightning detection system is considered the most accurate in the world by the WMO (World Meteorological Organization) (SM 2) for data pertaining to Western Europe, with an assured accuracy higher than 98%.

Following Table 1 shows the statistical results from seven stations based on the data provided by METEORAGE.

**Table 1.** Probability of lightning impacts in the area and incidence on infrastructure protected with ECCDs in seven geography points during large observation time.

| Lightning Impacts | LA MASELLA | LES PARDINES | AEMET JATA | PIC DE GOMÀ | ROC DE QUER | JAIZKI-BEL | ORDUÑA |
|---|---|---|---|---|---|---|---|
| Time period | 2014–2018 | 2003–2018 | 2010–2018 | 2015–2018 | 2016–2018 | 2015–2018 | 2015–2018 |
| On a 2 km around | 1.938 | 1.609 | 430 | 1.546 | 482 | 591 | 316 |
| On the tower (0 m) | 0% | 0% | 0% | 0% | 0% | 0% | 0% |
| Closer than 100 m | 0.2% | 0.12% * | 0.7% | 0% | 0% | 0% | 0% |
| At a distance between 100 m and 300 m | 5.37% | 1.74% | 4.65% | 0.39% | 1.87% | 0.85% | 0.95% |
| At a distance between 300 m and 500 m | 3.46% | 2.36% | 3.49% | 1.55% | 3.73% | 3,04% | 3.80% |
| At a distance between 500 m and 900 m | 16.56% | 9.82% | 22.56% | 6,47% | 14.11% | 17.09% | 16.14% |
| At a distance bigger than 900 m | 74.40% | 85.95% | 68.60% | 91.59% | 80.29% | 79.02% | 79.11% |

* This date has correspondence with two events: one in air cloud-cloud; the other an impact to 43.73 m from the tower on an electric media tension line (data: 14 June 2010 at time: 14:17:36, estimated electric current: 22 kA). The tower has not perturbed.

Figure 2 presents Les Pardines Tower in Eats Pyreenes (Andorra), as an example of a place protected with ECCD. It is one of those of included in the METEORAGE statistical study. In this case, Les Pardines point has the longest tracking period of analysis: 16 years (2003–2018).

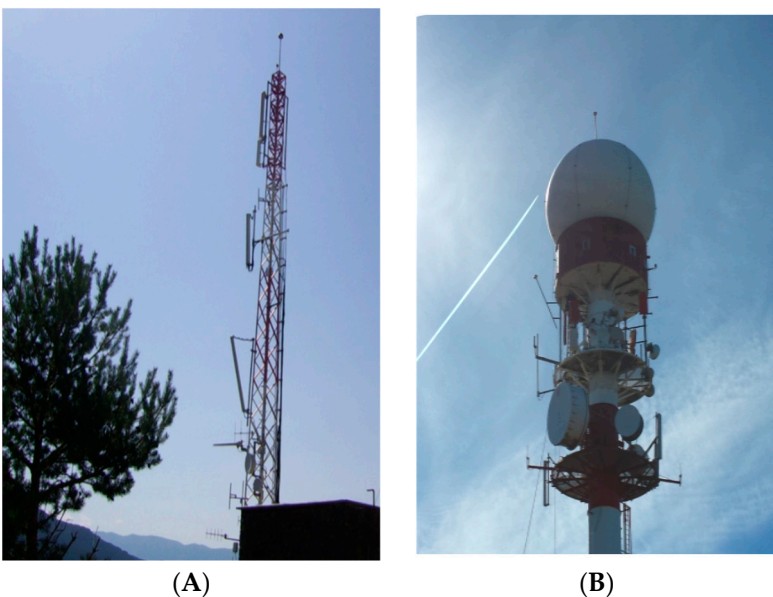

(A)            (B)

**Figure 2.** (**A**) ECCD 100 installed in the telecommunications tower Les Pardines in East Pyreenes (Andorra); (**B**) ECCD 100 installed in AEMET Meteorological Radar (Jata Tower) in Vizcaya (Spain).

The values in Table 1 allow us to conclude that the percentage of direct lightning impacts on the ECCD or the structure they protect is 0%, and within the maximum coverage radius of 100 m defined for the DDCE, the figure is 0.1%, increasing the probability of lightning impact as we move away from the influence of the ECCD. These data are especially relevant, since they are studies carried out over long periods of monitoring—16 years in the case of the Les Pardines Tower and 9 years in the case of the Jata AEMET Tower—and which, moreover, have always been carried out in areas with very high lightning density. When installing an ECCD, it is mandatory to ensure that all metal parts of the structure are protected, and at the same electrical potential as the lower hemisphere of the ECCD. This means that the influence of the ECCD will encompass the entire structure protected. Any metal structure or part that does not meet this condition will be out of its scope or protected volume.

Figure 3 shows a concentric diagram of protection efficiency, where the efficiency is considered as: 100 minus the probability in percent to receive a lightning discharge in each zone, relative to the distance where the ECCD is installed. The values are an average of the values obtained in seven tower stations. With an error <8/10.000 the protection area of an ECCD can be considered a semi-sphere of 100 m for the radius, but the coverage protection could be extended to a radius of 200 m without risk in recent installations (<4 years old), where the ECCD are more technologically advanced.

The scope of protection of an ECCD is basic depending on how far it can cause electrical discharges to be achieved by its influence. However, it is clear that an ECCD causes discharges in a quasi-continuous way. When a discharge take place, it is done consecutively through a new charge capturing new charges, their recombination, the compensation of electromagnetic fields, and the repeated cycles of those processes. Therefore, as the influence of the ECCD is sustained over the time, with a quasi-continuous operation, given that once the charges are compensated in a nearby environment at a given moment, the ECCD will be recharged, and repeatedly, the ECCD will re-compensate them, and so on. It seems intuitive that its influence on the existing charges in its environment will be much greater than 10 m and that, in any case, it will keep a relationship on the condition that all the existing conductive structures in its environment are at the same potential. However, for each time of discharge or when a charge compensation occurs, there is a withdrawal of mass in the volume of the environment of the ECCD that

unbalances the composition of the medium that surrounds the ECCD in the next volume. This will deal with the external forces that determine the internal dynamics of that environment: pressure gradients, local electromagnetic forces, forces linked to the viscosity of the medium, and the gravitational force [2]. All these external forces, and others linked to possible interactions of inter-atomic and inter-ionic forces, affect the gaseous diffusion processes of the ECCD's environment, towards an energy balance that could be considered adiabatic and isotropic, and isothermal processes that tend to homogenize the near environment—altered in a quasi-constant way by the performance of the ECCD. This will affect an extended environment in a way that is estimated to coincide with that which has been statistically well proven in these seven tower station points, assigning a secured minimum range of radio-protection efficiency of about 100 m for an ECCD: DDCE 100 model.

An ECCD is a compensator device of electrical charges in a nearby environment, and it collects pulses of electric currents associated with the cloud–cloud lightning or the other exogenous agents (including the eventual ground propagation of electric derivative pulses as a consequence of a closed or other, far-way lightning impact). At the same time, these electric pulses are a source of radio frequency signals, but they are not the only source; other causes may contribute to electromagnetic radio frequency, such as cloud movement, wireless communication signals, and variable wind friction. The electromagnetic radiofrequencies present in the environment of infrastructure protected by an ECCD are assorted. The property of the ECCD of being a local sink for radio frequency fields has been recently (March 2019) proven with an experimental real field test achieved by Dinnteco at the quay of Nagoya United Container Terminal Co. Ltd. in Tomihama, Tomi-shi, Aichi, Japan [3], where a large dock crane was being protected with an ECCD (DDCE 100 Plus model). Before using the ECCD protection, the NHK telecommunication tower near the pier generated serious discomfort in workers of a large dock crane, where the electric potential varies between 50 and 900 V (see register measures in Figure 4 A) but 1200 V peaks were measured as well. In this test, measurements of the electric current were from 5.31 to 39.22 mA. After the installation of the ECCD protection, the electric potential in equal points measured was reduce from 70 to 50 V (at same time electric current: a minimum of 0.94 mA and a maximum of 10.94 mA). All that was done without affecting the communications services established by the NHK tower.

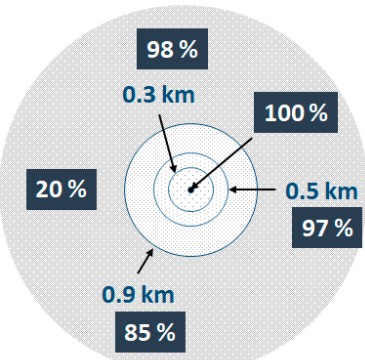

**Figure 3.** Diagram of the protection efficiency of an ECCD, model DDCE 100, as an average of statistical results obtained over long period time in seven different geographical points. Each diameter representing the radial distance to the ECCD is the efficiency, considered as difference to 100 minus the probability (in %) to receive a lightning discharge, in relation to installed ECCD position.

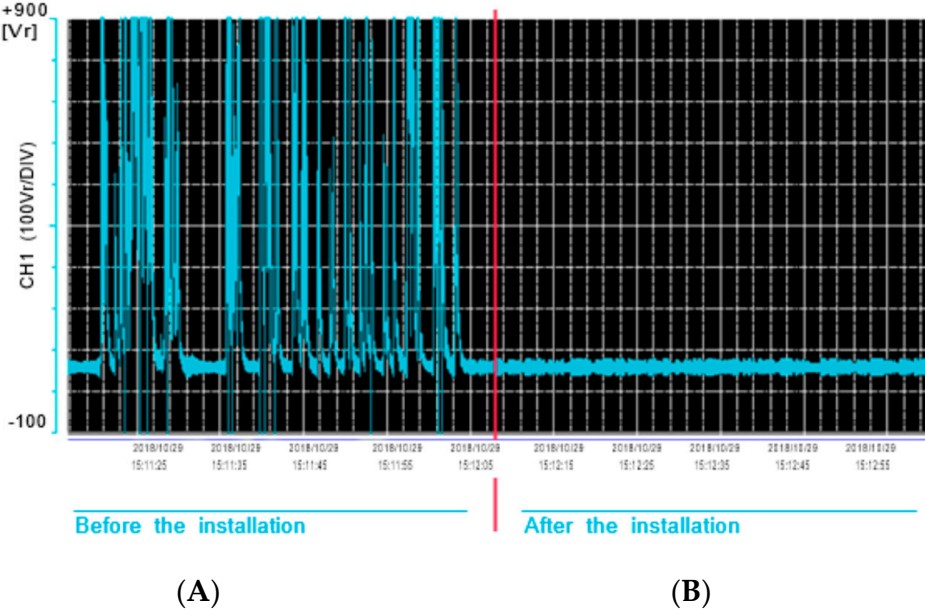

**Figure 4.** One of the register measurements of electric potential done at four different points of the large dock crane in Tomihama quay: (**A**) without ECCD protection; (**B**) with ECCD protection.

## 3. Simulation Study of the Electromagnetic Radiofrequency Field in an ECCD

### 3.1. Resonant Modes

A simulation study has been carried out with the help of the Radiofrequency Department of INTA—Instituto Nacional de Técnica Aeroespacial—using the Ansys HFSS software tool (SM 3), to evaluate the ECCDs' impacts on electromagnetic radiofrequency fields. The characterized devices correspond with the DDCE 100 and DDCE 50 models of ECCDs from DINNTECO; however, in this paper, we only present the results of the most general model—DDCE 100. In Ref. [4], the full report of both models is presented.

For the ECCD with the material properties and structural dimensions of the DINNTECO model DDCE 100, the simulation study obtained 40 different modes in the frequency range from 0.59 to 2.42 GHz. Outside that interval, no resonance modes were found (See Appendix A).

Figure 5 is a graphical representation to show in an intuitive way the ECCD resonance modes of DINNTECO DCCE 100, where the quality factor Q of each mode is represented against its frequency.

However, the resonance modes with low quality factor Q modes, <240–250, can be considered overlap. That is because of the fact that the simulation analysis did not take into account manufacturing dimensional tolerances of the devices. It did not consider either assembly deviations and dimensional changes due to thermal variation (expansions and compressions) that may contribute to uncertainties >3%, which correspond to random errors that are specified in the product of the convolution of the spectral profile of each mode by a Gaussian error function [5,6]. The consequence is an increase in the bandwidth of all modes with lower values in Q by a factor of 3 to 5. It is, therefore, more representative to consider that the performance of the ECCD with respect to radio frequency resonances consist of a continuous background and of two generalized main resonant modes (which are multi-mode envelopes with high values in Q). Figure 6 shows these considerations, and we have shown two asymmetric profiles (Fano type [7]) as two corresponding resonances with a quasi-continuous background of modes. In general, this quasi-continuous background is associated to a dissipative process.

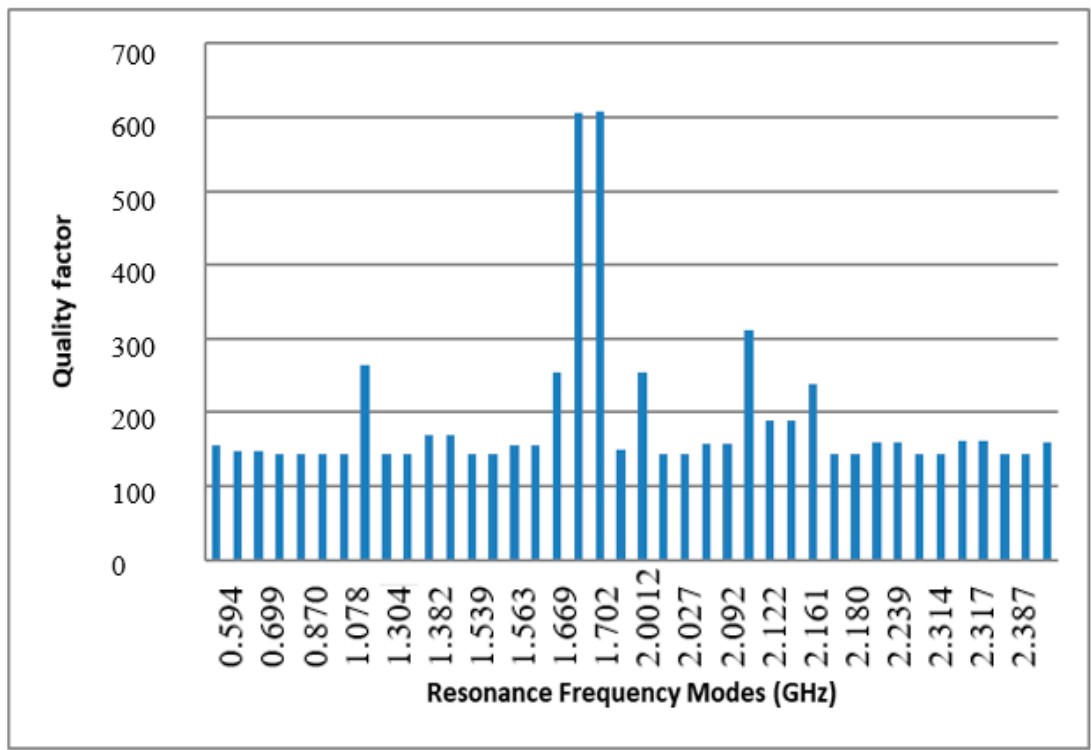

**Figure 5.** Resonance frequency modes of an ECCD corresponding to a DINNTECO DDCE 100 model.

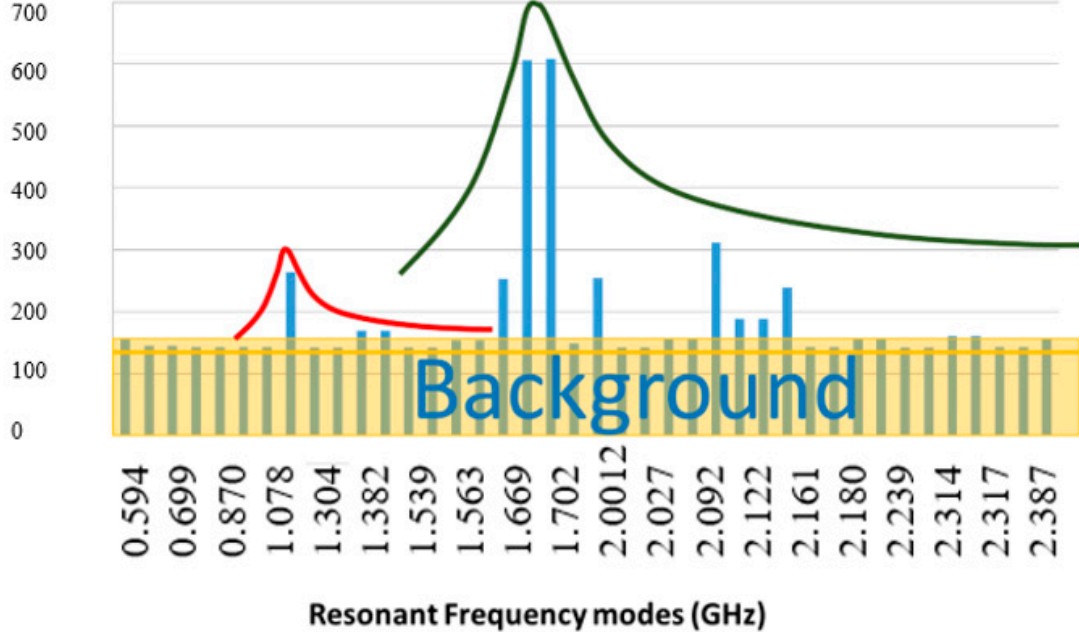

**Figure 6.** Functional behavior of resonant modes in an ECCD (DINNTECO DDCE 100 model).

Table 2 presents a numerical summary of the general characteristics, in terms of modal behavior of the DDCE devices, in which the Qeq value (equivalent Q factor) is included as an estimate for the background of modes, together with the characteristic values of the main mode. In the case of the DINNTECO DDCE 100 model, a second main mode is to the left (at 1.153 GHz) with a lower quality factor. We considered it more convenient to include in Table 2, as the main mode, the highest value of Q factor (which the simulation grants, in fact, to two overlapping modes and others with lower quality factors).

**Table 2.** General characteristics.

| EM FIELD | Main Mode | Mode Background | Factor Q in Main Mode | Factor Qeq Background |
|---|---|---|---|---|
| Attenuation time | 1.4 μs | 5.5 μs | ≈650 | 4.6 |
| Bandwidth | ≈2.8 MHz | 330 MHz | | 350 MHz |

The values that the simulation gives to the frequency of the main modes are quite in agreement with the dimensional geometry of the DINNTECO DDCEn 100 model. Thus, this gives us an approximate solution: using a fundamental equation applicable to the case of flat resonators that respond to the closing equation

$$\frac{y}{c} 2nd = 1,\tag{1}$$

where $d$ is the separation between electrodes (ideally: flat plates; real: caps), $y$ is the frequency of the mode considered, and $c$ is the speed of light in a vacuum ($2.9979 \times 10^8$ m/s). Therefore, for the mode selected as the main one in the DINNTECO DDCE 100 model, the deviations from the above-mentioned closing equation are +/− 6%.

The background attribution can be justified within a simplified form in the one-dimensional geometry of the DINTECO DDCE 100 model, with superior orders of non-axial resonant modes, specifically with orders 2 and 3 respectively. The closing Equation (1) used for the main modes was used for order 1. This seems correct for two reasons: the Qeq quality factor assigned is very low—which is compatible with separations between oblique electrodes—and because it is supported by simulation distributions of the electric and magnetic fields of the different modes analyzed; this will be used by us later. In any case, the geometrical dimensional correspondence coincides with the mean deviation value <10%.

Finally, the propagation time—commonly referred as "pulse duration", of the radio frequency fields in ECCD, can be estimated from the profiles of the modes, where their base frequency behaves as a radiofrequency wave carrier, and the rest of the profile as a radio frequency pulse shaper, which is how these radio frequency electromagnetic fields behave in practice. Without entering into a formal analysis, an estimate of the temporal duration of the pulse can be obtained by using the Fourier transform of the spectral power distribution that facilitates a propagation time [8], which in our case originates from a product of convolution of the base profile with a Gaussian error function, specified as

$$\text{T} \approx 0.7/\Delta y,\tag{2}$$

where $\Delta y$ stands for the bandwidth. With Equation (2), two propagation time ranges were obtained: 0.25 μs for the main modes and 2 ns for the continuous background of modes for the DINNTECO DDCE 100 model. The important point here is that in the ECCD, there are two propagation time ranges for radiofrequency electromagnetic fields.

### 3.2. Distributions of Electric and Magnetic Radiofrequency Fields, and Surface Current Density in an ECCD

In most cases, the simulation results carried correspond to timed records (videos: simulated image records) relating to spatial distributions in 2Ds and 3Ds with different spatial orientations of electric and magnetic fields (scalar intensity values and vector orientations), Poyting vector, and surface current density obtained in different chosen frequencies of the large spectral resonance of the ECCD in the DINNTECO DDCE 100 model (SM 1).

As a representative and significant example, we selected results of the radiofrequency field in the upper cap of a DINNTECO DDCE 100 model: electric and magnetic fields and surface current density at the resonant frequency 1.70 GHz for moments of temporal change characterized by the value of the phase in a semi-period. Five instances were selected at a phase step of 40° (see Table 3).

**Table 3.** Electric and magnetic fields and surface current density at the resonant frequency 1.70 GHz.

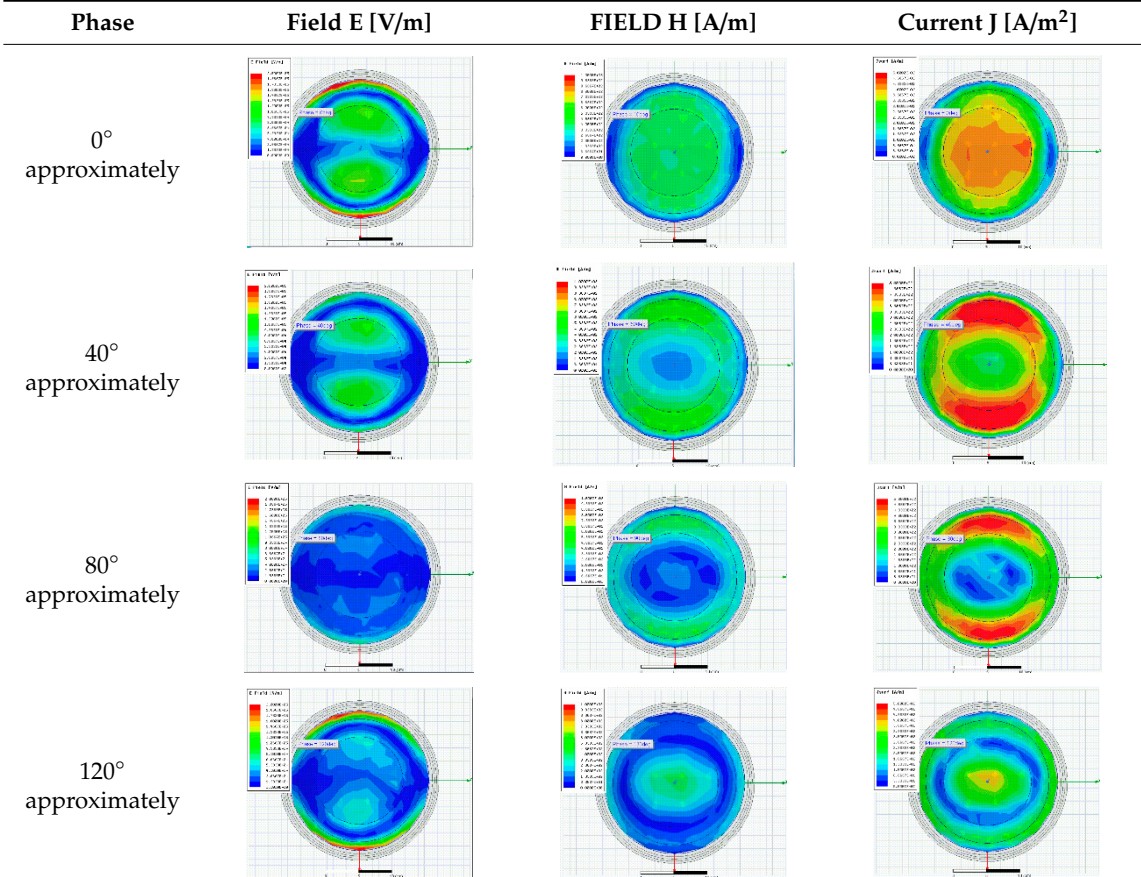

| Phase | Field E [V/m] | FIELD H [A/m] | Current J [A/m$^2$] |
|-------|---------------|---------------|---------------------|
| 0° approximately | | | |
| 40° approximately | | | |
| 80° approximately | | | |
| 120° approximately | | | |

A confinement in 2D and cylindrical revolution symmetry—very much in line with the designs of ECCD—is a situation analogous to the flat resonator case in 1D. However, the functions that represent the behavior of the electromagnetic field are somewhat more complicated. The mathematical expressions incorporate Bessel functions for the description of the confined resonant electromagnetic field. Additionally, the external part adjacent to the confined electromagnetic field (i.e., the evanescent electromagnetic field) continues to respond to a dampened exponential dependence, but is wider in scope regarding the volume.

This situation is considered a 3D confinement that will exhibit cylindrical symmetry. Additionally, some spatial composition with 3D symmetry—spherical or quasi-spherical symmetry—means that the qualitative behaviors of the electromagnetic field remain analogous to its resonant confinement. It will also mean the existence of an evanescent electromagnetic field. The analytical mathematical description in this 3D case is further complicated, as we need to resort to vector spherical harmonic functions that are dependent on Bessel functions and Legendre Polynomials [9]. But in the 2D case of cylindrical revolution symmetry, the analogy persists and the external field is evanescent and exponential in character. By breaking the 2D symmetry when passing to 3D in quasi-spherical conditions, the evanescent electromagnetic field will present azimuthal dependence; i.e., it will vary according to the angle of the direction of the electromagnetic field with the axis of revolution in 2D—axis OZ. All those reasons justify resorting to numerical solutions based on finite element calculations, which allows the results to be obtained faster and to be in accordance with geometric structures that are more complicated but more realistic. However, it should be noted that both the calculation tools used in the simulation and the analytical solutions which we have expressly mentioned in the fields of analysis—1D planes, cylindrical 2D symmetry, and spherical or quasi spherical 3D symmetry—are based on the Maxwell equations, and more directly, on the well-known Helmholtz spatial equation.

Therefore, it is crucial to access the interpretation of the results arising via simulation based on the support that can be found in knowledge of the different possible analytical solutions in ideal fields close to the geometric structures of real resonators, such as the ECCD case. This is because even the "language" designating modes, quality factor, resonant frequency, etc., are concepts derived from the analytical study of the different existing ideal models (1D, 2D, and 3D) based on the accessible symmetry they present.

Following the analysis of the simulation results, it can be concluded that:

- The ECCD—DINTECO DDCE 100 model—is an electromagnetic field resonant structure that has a main resonant mode at 1.7 GHz and a quasi-continuous oblique mode from 0.549 to 2.42 GHz. The main mode corresponds effectively to the central dimension in the intra-cavity axis of d = 83.79 mm in the so-called order mode m = 1 (called half-wave, $\lambda/2 = d$, empty). The prominent secondary or lateral mode at 1.153 GHz seems attributable to a confocal structure of the non-axial lateral parts that would be associated with a $4dn \approx \lambda$ (considering two straight sections: round-trip and two slightly oblique). The confocal condition is satisfied and maintained in the structures despite severe variations due to dimensional changes (derived from strong temperature variations) or changes in the refractive index of the medium. They can undoubtedly occur with a low-density plasma, such as the one produced during discharge by breakage of the dielectric rigidity inside the ECCD or the external cavity's resonant absorption. The secondary mode, therefore, acts as an external discharge insurance due of the strong evanescent field in the mode in relation to exogenous or endogenous external circumstances that can affect the ECCD.
- The quasi-continuous background modes are clearly angular modes, and they have an important contribution to external evanescent fields.
- Figure 7 shows results of the simulations.

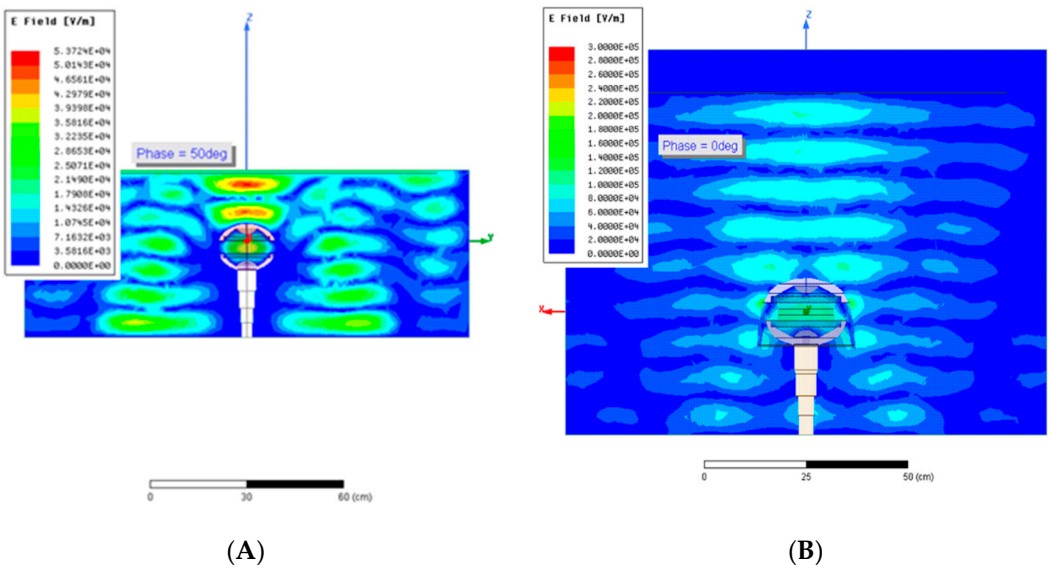

(**A**)                    (**B**)

**Figure 7.** Distribution of the radiofrequency electric field (V/m): (**A**) at 1.153 GHz and (**B**) at 1.71 GHz, both in the nearby free space in an ECCD—DINNTECO DDCE 100 model.

Figure 8 show simulations results obtained for the vector electric field (V/m) in an ECCD-DINNTECO DDCE 100 model: (A) 3D representation, (B) YZ plane cut, and (C) XZ plane cut. All of them for 1.153 GHz.

The secondary lateral mode at 1.153 GHz for the ECCD—DINNTECO DDCE 100 model—uniquely presents an intense outdoor electric field, but with a limited range (<50 cm radius) normal (X, Y) to the vertical (Z) axis. This mode corresponds to a "confocal" resonator scheme that works at approximately $4d \approx \lambda/2$. All modes, but most substantially the other modes that make up the quasi-continuous

background, contribute to the outdoor field with an equivalent but less intense range, although many modes make up the quasi-continuous background. This radiofrequency field forms a ring—a flattened toroid with an almost rectangular section (5–6 × 12–14 cm for the ECCD—DINNTECO DDCE 100 model). This ring surrounds the DDCE externally by the lower cap, and its average circumference corresponds with a resonance of the radiofrequency field of a high order, 12 to 14 times the wavelength. It reminds us of the so-called "whispering gallery" resonators [10]. The distance corresponds to half the length of a pulse travelling from the main modes (≅7.5 m), while the continuous bottom with the pulse length being 0.8 m for ECCD—DINNTECO DDCE 100 model. This results in compliance with the condition of transient stationery due to the possible interference of the pulse itself (given the length of the pulse, which evens overlaps once with itself), may be an indicator of the distributed functionality of each set of modal distributions and their dimensional incidence in ECCD performances.

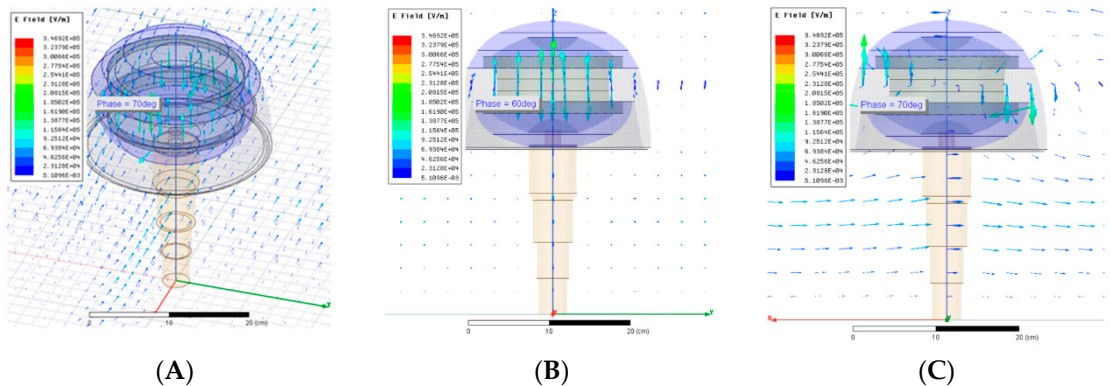

(**A**)    (**B**)    (**C**)

**Figure 8.** Simulations obtained from the vector electric field (V/m) in an ECCD-DINNTECO DDCE 100 model: (**A**) 3D representation, (**B**) YZ plane cut, and (**C**) XZ plane cut. All of them for 1.153 GHz.

Radiofrequency fields—as they are oscillating at high frequency—do not induce charge shifts, but—as they are resonant, and therefore stationary—they cause retention or entrapment of electric charges, which increase the effective recombination section to enhance charges compensation. Our working hypothesis is that the upper toroid ring traps and confines electrons or eventual negative ions. And the largest and farthest toroid ring, close to the lower caps and even to the stem, traps and confines positive charges (ions +), or most probable holes (positive electron charges). Due the fact that the positive ions have bigger masses, they will have difficulties for migrating from the earth to the lower caps. In Figure 9 is a representative presentation of this on an ECCD—the DINNTECO DDCE 100 model—photography.

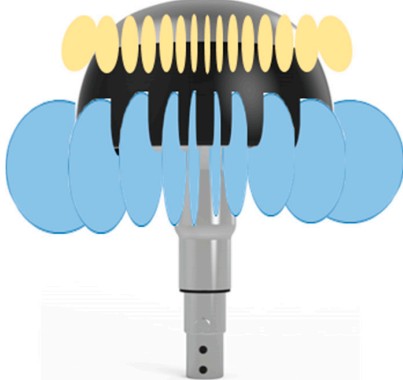

**Figure 9.** Graphical representation of spatial distributions of confined and intense evanescent field zones, corresponding to possible confinement of negative charges (in beige) and positive charges or holes (in blue).

## 4. Operation of an ECCD on a Tentative Quantum Scheme

In Section 2 of this paper, it was empirically proven that the ECCDs capture and compensate electrical charges (positive and negative) and that they strongly absorb radiofrequency (≈GHz). In Section 3, the numerical simulation analysis of radiofrequency field, whereby it is possible to infer the quality of multi-resonator behavior of an ECCD, was summarized. And at the same time, it was shown the spatial distribution of the evanescent field, mainly of electric field of radiofrequency, open the possibly confined electrical charges in two local zones, near to the environmental union of each ECCD caps with the dielectric isolator ring. Radiofrequency fields—as they are oscillating at high frequency—do not induce charge shifts, but—as they are resonant, and therefore stationary—they cause entrapment of electric charges, which increases the effective interaction among them and provides the recombination to enhance charge compensation. Our working hypothesis is that the upper toroid ring traps and confines electrons or eventual negative ions. And the largest and farthest toroid ring, close to the lower caps and even to the stem, traps and confines positive charges (ions +), or most probable holes (positive electron charges).

From the viewpoint of absorbable electromagnetic power, it is variable and depends mostly on the accumulation of charges over time and the resonance of the electromagnetic field frequency. In a simple scheme of the two levels [11], the physical statistics indicate that the thermodynamic equilibrium adjusts well enough to the distribution of Boltzmann populations, so that N−/N+ = 1.000221; that is, practically equal, so that any deviation of this equilibrium will tend to be compensated immediately by absorbed or stimulated emission process of electromagnetic radiation, that accompanies the recombination of electric charges. Without doubt, another absorption process will be responsible for the strong selective radiofrequency with a relatively large wide spectrum, in addition to the compensation of electrical charges.

All these results, and other coincidences, led us to consider the super-absorption process as a possible quantum-dissipative system.

An analogy may be made between the ECCD physical functioning and the quantum device through the super-absorption of light—SaQID [12], if the ECCD fulfills the requirements that the theoretical modeling imposes. It is true that the establishment of these requirements, as written in the quantum engineering [12], was orientated toward the nanotechnological field, mainly toward semiconductor quantum dots. Nevertheless, its background is in natural products extracted and modified from bio-synthetic materials with molecular ring structures. Meanwhile, the ECCD is a macro device (≈dm) absorbing or super absorbing radiofrequency fields ($\lambda \approx 10$ cm). In any case, the formal analogy may be established, not considering several orders of magnitude, provided:

1.  That the geometrical structure of both devices (SaQID and ECCD), the spatial distribution of its components (or parts), and the functions, are identical in both cases. This is clearly shown in Figure 1 of [12], transposed in Figure 10 of this paper, Figure 1 (ECCD model 100) and Figures 7–9 of this paper, together with the simulations of the electromagnetic field shown in Ref. [4].
2.  That the final behavior or both devices (SaQID and ECCD) is the same—strongly absorbing variable electromagnetic fields: light for SaQID and radiofrequency for ECCD. In general, both devices are strong energy dissipators for the capture of "excitons" (electrical charges of different signs and electromagnetic fields that are related).
3.  That the requirements of physical variables for a super-absorption process are fulfilled in an ECCD compared with the SaQID. This was proven, not considering several orders of magnitude, and with the available, reliable data.

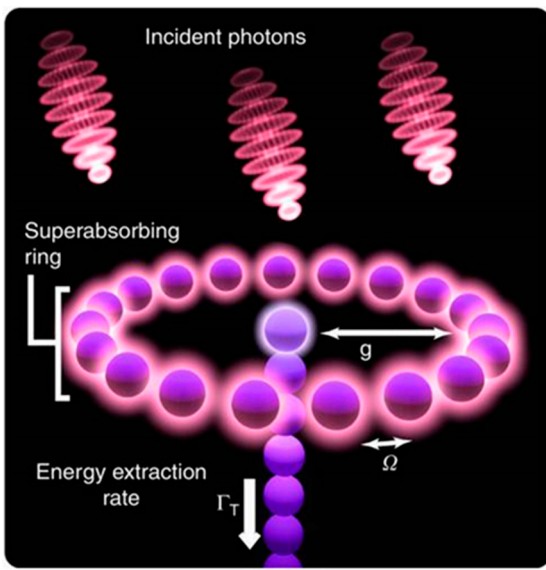

**Figure 10.** One potential realization of superabsorption [12].

The super-absorption process [12], as the reciprocal process of super-emission introduced 65 years ago by R.H. Dicke [13], happens over a set of N "quantum entities" that cooperatively interact with a surrounding electromagnetic field. In general, these quantum entities can be atoms, molecules, ions, semiconductor quantum dots, bio-synthetic molecular ring structures, etc. They have an allowed resonant and discrete dipolar transition. "Unlocated excitons" are also adequate quantum entities. An exciton may be visualized as a couple consisting of an electron and the associated hole, attracted to each other via Coulomb forces. Therefore, an exciton represents a quasi-particle or a "free exciton" in the Wannier exciton sense. A set of M electrons associated with M holes is also an exciton—an "exciton cluster" or "exciton cell". It can collaboratively interact with a surrounding electromagnetic field. A set of P "exciton clusters" could be available to collaboratively interact if the surrounding field has a spatial and temporal coherence—length of pulse—larger than the spatial dimension for the ubication of the P collaborative set of an "exciton cluster". In principle, these are the candidates to be considered able to adapt to a super-absorption process.

The ECCD works in environmental temperatures (5°–25 °C ≈ 278°–298 °K, average: 280 °K) and will be associated from statistic thermodynamic bases to a "binding energy" (or ionization energy) $E_x$ ≥ kT, where k is Boltzmann constant. In these conditions, the binding energy $E_x$ ≈ 8.3 $10^{-6}$ eV involves a singled exciton electron-hole, and the ratio between these two energies gives N; the approximated number of exciton clusters. M ≥ 3000. The value of binding energy $E_g$ is compatible with the Columbian energy $E_C$ of an exciton cluster, if distance r ≈ 16 cm and relative electric permeability $\varepsilon_r$ ≈ 2.4. This last value is realistic for a cold plasma in low frequencies (GHz)

$$E_C = (N_e)^2 / 4\pi\varepsilon_o\varepsilon_r. \tag{3}$$

A density of charges $N_e$, $N_h$ (electron and hole) can be derived from of Debye length considered in the order of micrometers, and is well-compatible with the take electric permeability value, $N_e$ ≈ 4 × $10^{10}$ m$^{-3}$.

These values show that squaring possible number P of exciton clusters in an ECCD (DDCE 100 model) wavelength of radiofrequency field P, must result in P < 20 = ring length/$\lambda$, and in consequence

$$N_e, N_h \geq N = P\,M. \tag{4}$$

There is an inefficiency of about two orders of magnitude in the exciton clusters' formation, but the values is possibly less in the single excitons' formation. In an ECCD, the excitons are

confined independently into the set of $N_e$ electrons and $N_h$ holes in specific areas around the device. The numerical simulation analysis of radiofrequency field—Section 3—showed the quality of multi-resonator behavior of an ECCD. At the same time, the spatial distribution of the evanescent field, mainly of electric field of radiofrequency, opens up the possibly confined electric charges to two local zones, near to the environmental union of each ECCD cap with a dielectric isolator ring. These confinements are not permanent but temporary: we estimated the total effective time in a temporary fraction $\approx 4/9$ of one period time, divided into two short times $\approx 2/9$ of one period time. In our case of DINNTECO ECCD 100, the short time of pulse confinements is $\approx 0.15$ ns with a cadence repetition of $\approx 0.65$ ns. The intervals of time-free out pulses' confinements are sufficient to assure the persistence in time of the electric charges, because the time of diffusion and displacement are the largest $\approx 10^{-4}$ s. The distance for the confinement rings was established to be 30–36 cm radially at the symmetry axis, in accordance with the dimensions established in Section 3. The existence of resonator conditions of an ECCD and the electric charges confined, gives the conditions necessary for an interaction among them. It is not sufficient: another factor that gives spatial mobility to electric charges in order to mix the two confined charge rings. Either way, the transit time between the two confined charge rings was estimated to be around 20 to 30 μs, considering a static electric field value between $3 \times 10^4$ and $10^5$ V/m, with the distance between the two rings being 10 to 30 cm. These recharges came from the superficial of current density (see value of Table 1, row Current **J**) and there are instances where the superficial density of current has two symmetrical peaks on the edge of the cap and the dielectric. The maximal peaks of superficial density of current are the vanishing points of electric charges. This facilitates, along with the bipolar charges, mobility: the temporary establishment of a potential current bridge.

The grounding of the ECCD's lower cap allows an electrical potential difference to be established with the charges in the device, but an ECCD is also an electromagnetic radio frequency resonator in the range of 0.50–2.20 GHz, which facilitates electrical current discharges (that is experimental evidence). Discharges take place by the absorption of radiofrequency electromagnetic fields and the recombination of electrical charges of different signs (mostly electrons and holes or ions +). This happens through the participative action of the static electric and magnetic fields, as well as the resonant electromagnetic field in several frequencies. Both actions facilitate the combination of charges when these are withheld—temporarily trapped—in nearby areas of confinement by the effect of the radiofrequency electromagnetic field, where the action of mobility promoted by the static fields enhances the compensation of the electric charge. Figure 11 is another simulation study prepared with Consol Multiphysics Module 1.1 AC/DC (SM 4) that obtains the electric static field of an ECCD. These processes take place in the immediate external surroundings of the ECCD (up to a maximum distance where electric static field is uniform is $\approx 10$ m), but their permanent and almost continuous activity is felt resonating in an environment of a greater range (up to > 200 m), because the compensatory process of homogenization promote the diffusion between media in unbalanced local environments.

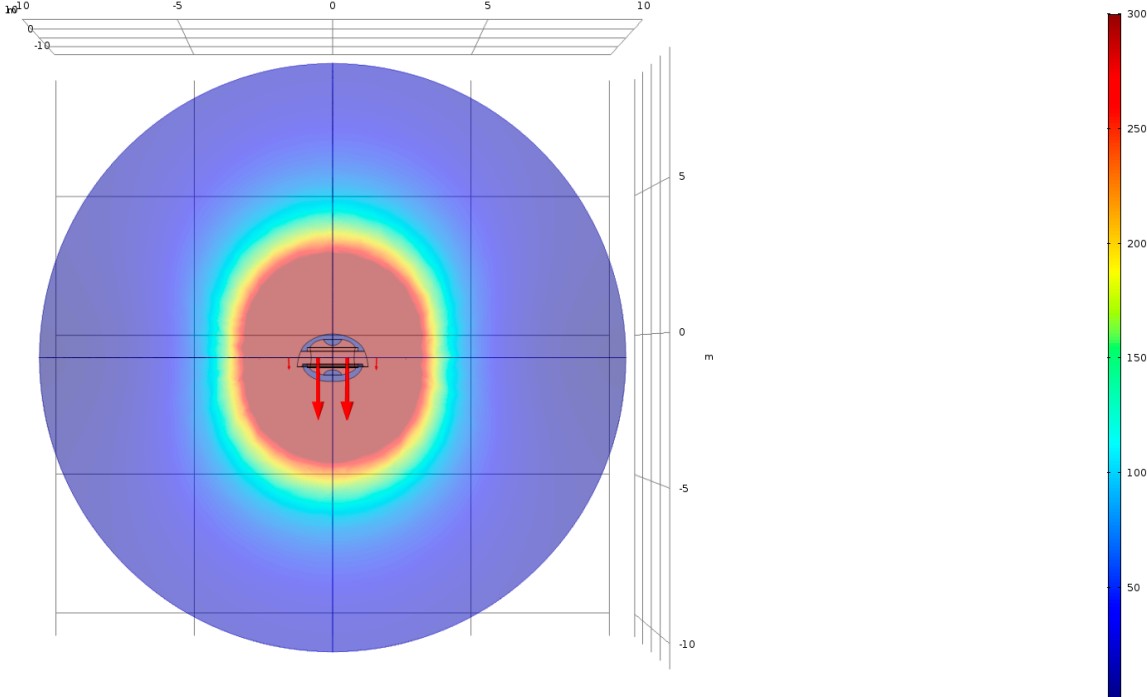

**Figure 11.** Electric static field of an ECCD—DINNTECO 100 model.

An ECCD has a principle of operation on the offset electric charges that exists in the surrounding environment, avoiding an ascending liner generated in the ECCD and absorbing electromagnetic radiofrequency energy in the environment of the protected structure. The compensation of electrical charges in an ECCD is a cooperation of variable electric fields (temporal evanescent confined radiofrequency) and electrostatic field surroundings of ECCD. But, an important question needs to be answered: How is it possible to extract the energy? This problem is present also in photovoltaics (PVs) and artificial photosynthetic systems based on excitons [14]. In fact, how the energy must be lost is in overcoming the Coulomb barrier, converting excitons into free charges, fundamentally, as it determines the achievable open-circuit voltage or over-potential. The ECCD has solved this serious technical difficulty using the offset time operation: the confinement is only maintained in time, only 4/9 of the fraction time of the period is operative in two fractions of 2/9 the time of period. The rest of the time, the electric compensated electro-hole pair charges are free but close to contact with earth connection, and Coulomb barrier is suppressed. For instance, the energy of ~1 eV is involved in the electrostatic field on an elemental charge at 10 μm of distance.

## 5. Conclusions and Future, Novel Applications

An empirical rule on the efficiency of an ECCD as security against impacts of lightning was obtained from an exhaustive statistical analysis of METEORAGE environmental services on lightning nearly and proximity (<2 km) of seven station point choices by "ceraunic" activity. For an ECCD—DINNTECO DDCE 100 model, the top efficiency (100%) of protection was considered a semi-sphere with a radius of 100 m. Other models with different dimensions and geometrical configurations may establish other empirical rules for the efficiency of protection and may change the role of an ECCD as an inhibitor of radiofrequency and a suppressor of pulsing electric current. Recently, an experimental proof has given evidence for the habituality of an ECCD to eliminate the radiofrequency perturbations induced by emissions of a near telecommunications tower on a piece of infrastructure—a large dock crane.

From a simulation of an electromagnetic radiofrequency field into and out of an ECCD's vicinity, in particular, for a DINNTECO DDCE 100 model, we can confirm that an ECCD has two principal resonant modes: one associated to central gap of electrode-caps, and the other, of less frequency,

to a confocal resonance configuration. This last resonant mode presents an important evanescent electric field. A large list of overlap modes justifies its consideration as a quasi-continuous background frequency with limited band bandwidth. All these incidences are 100% compatible with a Fano profile for the spectral resonance, where background is associated to dissipative process. The simulations also gave spatial distributions in 2Ds and 3Ds of electric and magnetic fields, and the other electric parameters—the Poynting vector and superficial electric current in the caps, for the different resonant radiofrequency modes. The analysis shows an important evanescent electric field for the less in-frequency resonant mode that promotes temporal electric charge confinement in two rings in the near vicinity of ECCD, each of them near of metal caps' dielectric union. The diffusion and buffer collision process is made quasi-permanent the electric discharges in the indicated zones of the ECCD.

The initial and nearly-permanent presence (during time of charge/discharge is kept) of a constant and uniform static electric field in ECCD's surroundings (<5 m), established, principally, pairs of electron-holes (ion–/ion+, and other negative/positive combination pairs are possible). The time duration pulse associated to resonant modes and a confined pair, deals with the possible process of super-absorption (proportional to $N^2$ excitons), as an exclusive process of dissipative energy in an ECCD. The finite time of radiofrequency establishes a temporary electrical current bridge on specific points, that facilitates the electric discharge that removes electric charges and energy associated to a radiofrequency field.

Recently, Dinnteco has been planning new research to adapt an ECCD's use to wind turbines' electric energy generators. This has dealt with a new configuration of ECCD and two new complementary products (SM 4) available in the market as advances in reduction of radiofrequency incidences on renewable energy sources in form of wind turbines. An increased tax on efficiency is expected.

Another possible ECCD use is concurrent use with adequate infrared sources, in order to increase the visibility on the road in smoggy conditions, cause by the presence of stratocumulus clouds.

Finally, an important task is to study the impacts of water's physicochemical properties—singularly and for the aggregate states of water—on the electric discharge times of installed ECCDs. It is a conceptual thesis that they could have relevant consequences, and they could be related to the possible cluster formation of the confined charges in an ECCD. Both phenomena could be responsible at the terminus of electric current discharge on the ground with a duration time of approximately ms.

**Supplementary Materials:** The following are available online at http://www.mdpi.com/2076-3417/9/22/4879/s1, SM 1. www.dinntecointernational.com; SM 2. https://www.meteorage.com/f; SM 3. ANSYS: HFSS: https://www.ansys.com/products/electronics/ansys-hfss; SM 3. CONSOL MultiPhysics: https://www.comsol.com/; SM 4. New Dinnteco complementary products: DINFIL and DNNFT, both two model of radiofrequency filters; DINEOL, a specific ECCD component for aero generators.

**Author Contributions:** Conceptualization, E.B. and M.A.-S.N.; methodology, E.B. and M.A.-S.N.; software, E.B.; validation, E.B., J.M., and M.A.-S.N.; formal analysis, E.B. and M.A.-S.N.; investigation, E.B. and M.A.-S.N.; resources, E.B. and J.M.; data curation, E.B., J.M., and M.A.-S.N.; writing—original draft preparation, E.B., J.M., and M.A.-S.N.; writing—review and editing, E.B., J.M., and M.A.-S.N.; visualization, E.B. and M.A.-S.N.; supervision, E.B.; project administration, E.B. and J.M.; funding acquisition, J.M.

**Funding:** This reseach received no external funding.

**Acknowledgments:** We thank Borja Plaza-Gallardo for his excellent work and David Poyatos-Martínez and Daniel López-Sanz for assistance with the numerical ANSI simulations at the Instituto de Tecnología Aeroespacial INTA; and A Hala Kamal Abd El Hady, from the Ain Shams University, Cairo (Egypt), for her contributions to the numerical simulations and support with Consol Physics.

**Conflicts of Interest:** The authors declare no conflict of interest.

**Patens:** P1. Dinnteco International: Spain Reference: P201530389. Starting date: 24.03.2015. Concession date: 05.02.2016; USA Reference: 9.685.775 B2 (14/985 708). Starting date: 31.12.2015. Concession date: 20.06.2017. In other 33 countries with concession or starting dates.

## Appendix A

**Table A1.** Resonant modes in an ECCD (DINNTECO DDCE 100model).

| Frequency (GHz) | Damping (GHz) | Q | Bandwidth (MHz) | Comments |
|---|---|---|---|---|
| 0.594 | 0.0019 | 155.662 | 3.813 | |
| 0.699 | 0.0024 | 146.399 | 4.772 | 0 |
| 0.699 | 0.0024 | 146.393 | 4.773 | Overlap |
| 0.870 | 0.0030 | 144.099 | 6.038 | 0 |
| 0.870 | 0.0030 | 144.075 | 6.0413 | Overlap |
| 1.077 | 0.0038 | 143.244 | 7.521 | 0 |
| 1.078 | 0.0038 | 143.221 | 7.530 | Overlap |
| 1.153 | 0.0022 | 264.885 | 4.354 | 0 |
| 1.304 | 0.0046 | 142.920 | 9.122 | 0 |
| 1.304 | 0.0046 | 142.914 | 9.124 | Overlap |
| 1.382 | 0.0041 | 169.699 | 8.143 | 0 |
| 1.384 | 0.0041 | 169.995 | 8.140 | Overlap |
| 1.539 | 0.0054 | 142.869 | 10.771 | 0 |
| 1.540 | 0.0054 | 142.810 | 10.787 | Overlap |
| 1.563 | 0.0050 | 155.189 | 10.071 | 0 |
| 1.564 | 0.0050 | 155.113 | 10.082 | Overlap |
| 1.669 | 0.0033 | 253.406 | 6.588 | 0 |
| 1.701 | 0.0014 | 605.818 | 2.807 | 0 |
| 1.702 | 0.0014 | 608.044 | 2.799 | Overlap |
| 1.756 | 0.0059 | 149.011 | 11.785 | 0 |
| 2.0012 | 0.0039 | 254 | 7.873 | |
| 2.0253 | 0.0071 | 143 | 14.183 | 0 |
| 2.0270 | 0.0071 | 143 | 14.196 | |
| 2.0927 | 0.0067 | 157 | 13.300 | 0 |
| 2.0929 | 0.0067 | 157 | 13.321 | Overlap |
| 2.0958 | 0.0034 | 312 | 6.716 | Overlap |
| 2.1222 | 0.0056 | 189 | 11.249 | 0 |
| 2.1222 | 0.0056 | 189 | 11.249 | Overlap |
| 2.1614 | 0.0045 | 239 | 9.052 | 0 |
| 2.1799 | 0.0076 | 144 | 15.126 | Overlap |
| 2.1800 | 0.0076 | 144 | 15.127 | Overlap |
| 2.2386 | 0.0071 | 158 | 14.130 | 0 |
| 2.2386 | 0.0071 | 158 | 14.125 | Overlap |
| 2.2391 | 0.0078 | 143 | 15.672 | Overlap |
| 2.3139 | 0.0081 | 143 | 16.204 | 0 |
| 2.3171 | 0.0072 | 162 | 14.320 | Overlap |
| 2.3173 | 0.0072 | 162 | 14.322 | Overlap |
| 2.3870 | 0.0083 | 143 | 16.649 | 0 |
| 2.3873 | 0.0083 | 143 | 16.651 | Overlap |
| 2.4177 | 0.0076 | 158 | 15.280 | Overlap |

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
