# Peer review of "An ECCD—Electronic Charge Compensation Device—As a Quantum Dissipative System"

_applsci, doi:10.3390/app9224879_

Round 1

Reviewer 1 Report

I have the following objection to the paper

In my opinion the experimental results are not conclusive in the way that has been presented in the paper. So the authors claim that  the system avoids the  Lightning Impacts, however in LES PARDINES, in the time period 2003-2018 there were on a 2 km around the antenna 1.609 Lightning Impacts. That means 0.00013 Lightning Impacts/m2. Then, with a simple calculus in 100m around tower, I would expect about 4 Lightning Impacts with no device and they detect 2 that is within the statistical fluctuations. 

In may opinion a conclusive experiment will be 

a) Detect how many Lightning Impacts there are in the tower without the system during an adequate period of time b) Detect how many Lightning Impacts there are in the tower with  the system during the same period of time c) The grounding an the shape modifications are very important in these kind of experiments so it is also important to detect how many Lightning Impacts there are in the tower with the whole  system except the  DINNTECO DDCE 100 device during the  same period of time. It will be even better to put something with the same shape instead of the device

The experimental results performed with the crane have the same problem I would expect results with and without device head.

These objections can be solved prior to considered the paper acceptation  

Author Response

The telecommunications towers, meteorogical stations and infrastructure of any other informative nature (traffic control, fauna paths …) used to be placed in the higher points of the orography. If they are provided with electrical and electro-technical parts, they have to fulfill the Directive on electro-technical equipment CEE 98/34 CEI-CENELEC (ES: RD 842/2002, adopted in 08.02.2002 and revised in 04.06.2019). In this directive is mandatory to have the equipment connected to an electrical earth with an electrical resistance below 10 ohms. In the 7 towers referred in the paper together with the Tomihama Quay crane (Japan), the earth connections are below 5 ohms. This means that a point with a voltage equal to the electrical earth in the high point, has a big probability to occur a lightning around it. Therefore, those installations should be protected. The electrical protection offered by the Franklin lightning rod causes an increase in the probability of a lightning impact. Its discharge should be derived to earth, nevertheless, it does not assure zero damages in the infrastructure and consequently in the electro-technical equipment. In fact, under these conditions, the tower Les Pardines, Andorra, has two severe lightning when it was about to install an ECCD 100. One of them caused a total destruction. The installation of an ECCD 100 has made almost disappeared the risks.
About the two lightning registrations recorded by the Meteorage at Les Pardines, it has to be precised that one of them was a cloud-cloud and the other (date:06.14.2010. time:14:17:36h. valued intensity current: 22kA ) took place at 48,73 m from the Les Pardines tower, on a media voltage power line . The tower Les Pardines was not damaged this time. It could be also possible that the lightning happened far from 48,3 m, but that it showed up at that distance due conduction through the high voltage power line. There is a call (*) in Table 1 of the paper, in order to take account of the Meteorage discriminaton of the register cloud-cloud and cloud-earth.
We include an informative annex about Les Pardines (Andorra) from the owner company of the communication tower (Andorra Telecom).
Before installing the ECCD in the towers AEMET/Jata (tower for data collection and transferring of information of the Spanish Agency of Meteorology) and the telecommunications one in Jaizquibel (Vizcaya) had quite dis-comfortable records, with many incidents and even severe damages. But table 1, that gathers in detail the Meteorage data, indicates 0 lightning for the 7 towers referred under the ECCD protection and in long temporal periods (continued for several years).
Figure 3 plots the radial dependence of the coverage of the protection: presented as Efficiency. It is stated that the consideration for the lightning/area has no interpretation because of the high increased probability of lightning in an infrastructure at a high altitude and equipotential to earth (resistance < 5ohms.
In the big port crane (test developed in November 2018) in the Tomihama Quay, Japan, the structure was connected to the electrical earth (electrical resistance < 5ohms) before installing the 4 ECCD 100 in well thought points. This spatial connection did allow voltage gradients of even 50 000 V, although of quite small intensity. This was the cause for headaches and general discomfort for crane operator, moreover, increasing the risk of potential cancers. Once the 4 ECCD 100 were installed, there was a voltage gradient not over the 50 V (see figure 4). Right now, there has been reported no headaches or discomfort among operators.
Finally, it is convenient to remark that DINNTECO has an agreement with METEORAGE only for following of the lightning data in surroundings and the geometrical points where the ECCD has been installed. This has a cost of 1 000 €/point and year. Therefore, this paper had to be restricted to 10 points by now: a telecommunications tower with results for its first year and two other points recently incorporated in Istria (Turkey).

Reviewer 2 Report

The authors have described the mechanism of an Electronic Charge Compensation Device. The device has been simulated and manufactured. Several EM measurements and in service statistic are presented. The interaction of the EM fields and the charge compensation match the experimental data of the device.

A revision of English is suggested, some sentences are quite confusing.

There are not any resonances above 2.42GHz? It seems strange a cut-off frequency on this structure. Maybe high Q resonances are not present. Please clarify this aspect.

Author Response

The numerical simulations developed carried out with HFSS software tool (by INTA)[SM3, 4] came up with a cut-off at high frequency (2.42 GHz) for DDCE 100 & DDCE50 models, but the cut-off at low frequency is different for each model: 0,594 GHz for DDCE 100 and 1,118 GHz for DDCE50 (see figure en PDF anexe:Graphical Mode distribution in a DDC50 model).

That is compatible with the geometrical structure of the ECCD cavity: the shortest distance between electrodes – upper and lower spherical cap- is around d = 60 mm in both models. The distance just in the nearby of the dielectric 2d about ƛ, where ƛ = 12 µm (~ 2.42 GHz), and the electric permissibility has to be considered 1 as there is vacuum inside the cavity. Nevertheless, the cut-off in low frequency (long wavelengths) is conditioned by the different radial structure and the curvature of the inner surfaces for both models. It is included a pdf file with the 2D geometrical configuration of model ECCD 100. The radial and curvature changes in the ECCD 100 with respect to the ECCD 50 oscillates among 1,25 and 1,13-1,40, respectively. Despite their similarity, they are different. This explains the modal distribution to be different. It can be established that the evanescent field is higher in a ECCD 50 (bigger Q also) than in the ECCD 100.
In the original manusript we have included only some data relative to the standard product ECCD 100, in order not to increase its length.

Reviewer 3 Report

I think that the last part of the section 4, at the page 12, has to be improved.

Here, the authors report an extra absorption of the device which cannot explained by theoretical simulations. 

They the hypothesis that this can be explained due to "collective effect" or super-absorption process studied in Ref.[11], which is defined as the inverse process of Dicke's super emission.

But, in Ref.[11], the authors discussed a specific model that was designed to reach super-absorption.

Why the ECCD device analyzed in this paper should encode a such theoretical model?

I think that the authors have to describe better what is the theoretical model that they have in mind and how it is connected with the device that they studied.

Author Response

We agree with Referee 3 about his suggestions on Section 4, page 12 and on, in order to improve it and make it more clear for the comprehension of the functional behavior of an ECCD. Therefore, we have rewritten this Section.
This new version has been built up presenting an analogy between the ECCD devices with Quantum Engineering by super absorption process -SaQI-, whose technological bases has been considered in the nanotechnology field, [11]. It is proposed to include another reference in order to improve the reading of the paper. The new version does establish a quite formal analogy well justified for defining an ECCD as a quantum dissipator of energy, as far as possible.

Round 2

Reviewer 3 Report

I have read the  revised manuscript. It can be found that the authors answered my queries and added paragraphs 
according to my suggestions. I have no further questions and recommend it to publish.